# A Comparative Analysis of Two Major Approaches for Mapping the Wildland-Urban Interface: A Case Study in California

**Avi Bar-Massada** 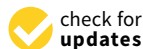

Department of Biology and Environment, University of Haifa—Oranim, Kiryat Tivon 36006, Israel; avi-b@sci.haifa.ac.il; Tel.: +972-4983-8923

**Abstract:** The Wildland Urban Interface (WUI) is where human settlements border or intermingle with undeveloped land, often with multiple detrimental consequences. Therefore, mapping the WUI is required in order to identify areas-at-risk. There are two main WUI mapping methods, the point-based approach and the zonal approach. Both differ in data requirements and may produce considerably different maps, yet they were never compared before. My objective was to systematically compare the point-based and the zonal-based WUI maps of California, and to test the efficacy of a new database of building locations in the context of WUI mapping. I assessed the spatial accuracy of the building database, and then compared the spatial patterns of WUI maps by estimating the effect of multiple ancillary variables on the amount of agreement between maps. I found that the building database is highly accurate and is suitable for WUI mapping. The point-based approach estimated a consistently larger WUI area across California compared to the zonal approach. The spatial correspondence between maps was low-to-moderate, and was significantly affected by building numbers and by their spatial arrangement. The discrepancy between WUI maps suggests that they are not directly comparable within and across landscapes, and that each WUI map should serve a distinct practical purpose.

**Keywords:** Wildland Urban Interface; buildings; wildfire; mapping

## 1. Introduction

The Wildland Urban Interface (WUI) is the area where human settlements meet or intermingle with natural or semi-natural ecosystems [1–5]. The WUI has grown in past decades, especially in the US [2,6,7], but also in other countries [8–11], exacerbating multiple environmental problems the emerge from interactions between human-dominated and natural systems. Due to historical reasons, WUI research has largely focused on environmental problems related to wildfire [12]. However, houses in the WUI also affect neighboring ecosystems through biotic processes including exotic species introduction, wildlife subsidization, disease transfer, land cover conversion, fragmentation, and habitat loss; and via abiotic processes such as wildfire ignition and spread, and soil, water, air, and light pollution [3,13]. This wide array of negative effects of buildings in the WUI, coupled with its large spatial footprint [1,14–16] and continued growth into natural and semi-natural ecosystems [2,6,7], highlight the need to increase the understanding of the spatial patterns of the WUI across broad geographical extents, as well as its effects on the biotic and abiotic components of landscapes. To achieve the former, we first need to gain a clear understanding of the effect of WUI mapping approaches on the spatial patterns of the WUI, and on the implications of these patterns [12]. This manuscript is motivated by this challenge.

At present, there are two main approaches for mapping the WUI, the zonal-based [1,2] and the point-based [14,17,18]; though other approaches based upon different WUI concepts exist as well [16,19]. Both the zonal-based and the point-based approaches are based on the same concept, which is outlined in Figure 1. A given spatial unit in a land cover map

is defined as WUI if building (or housing) density within it exceeds a certain threshold, and the percent cover of specific land cover types (flammable vegetation in the case of the fire-centric WUI) within it, or in a pre-defined area within a given distance to it, exceeds a specific threshold (Figure 1). In the standard US case [1,17], which focuses on wildfire risk, the building density threshold is 6.17 /km$^2$, and the flammable vegetation threshold is 50%. A further set of thresholds is used to differentiate between two sub-classes of WUI, interface (where settlements adjoin wildland ecosystems) and intermix (where settlements are interspersed within a matrix of wildland ecosystems). Intermix WUI is designated using the above thresholds, and interface WUI uses the following thresholds instead: less than 50% flammable vegetation inside the unit, but more than 75% flammable vegetation in another unit which is larger than 5 km$^2$, and within a distance of 2.4 km from the focal unit (Figure 1). These sets of thresholds reflect the designation of the WUI as an area where wildfire is the focal environmental problem, and they are parameterized accordingly. However, it is straightforward to re-parameterize these two approaches in order to generate WUI maps that reflect other environmental processes beside wildfire.

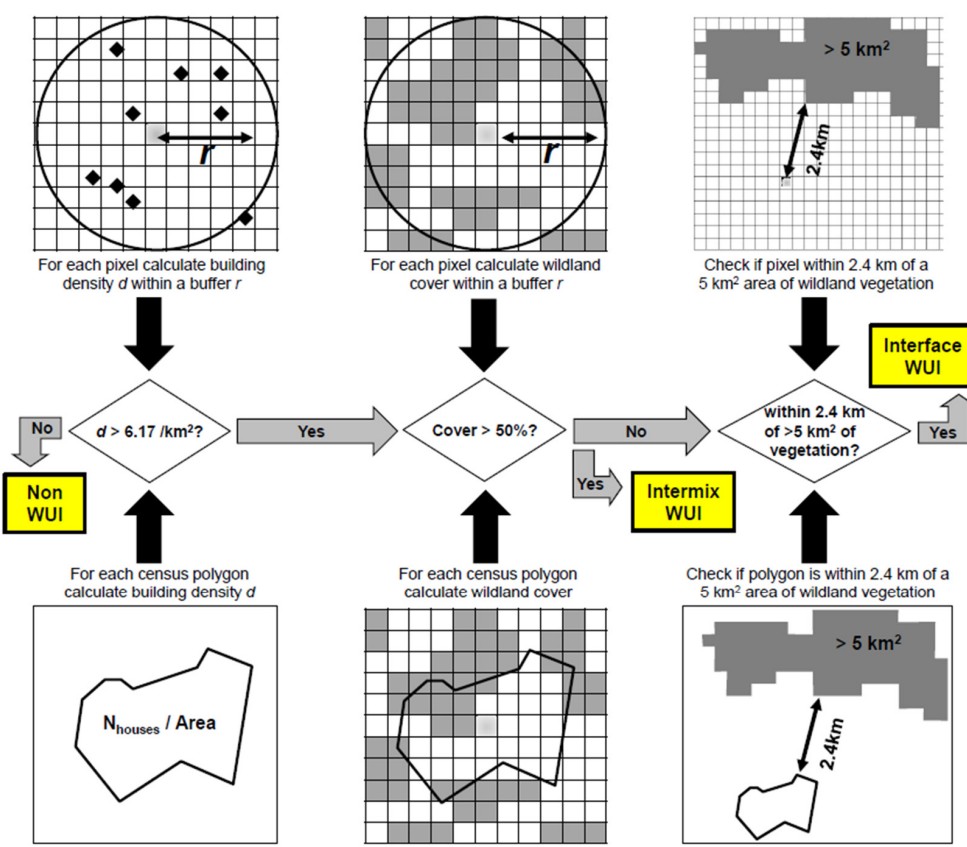

**Figure 1.** A schematic representation of the two WUI mapping approaches used in this study. Notice the decision processes in the middle are the same for both approaches. The top part of the figure was adapted from [17]. Glossary: intermix WUI is where buildings are interspersed within wildland vegetation; interface WUI is where clusters of buildings adjoin or border patches of wildland vegetation.

While the zonal-based and the point-based approaches for mapping the WUI share parts of their methodology, they differ in one thing: the type of geospatial data on building locations across the landscape, which serves as the input for the calculation of building density. In the zonal-based WUI, the starting point is a polygonal land cover map, in which

each polygon contains an attribute that reflects the number of buildings (or specifically houses) within it. The polygons, which typically vary in shape and size, serve as the fundamental spatial units of the subsequent WUI map. For example, the most commonly used WUI map in the US [7] is based on the US census data, which comprises census block polygons that can vary considerably in size and shape both within and across landscapes. In contrast, in the point-based WUI, there is no land cover map with information on housing to begin with. Rather, the fundamental data objects are points, reflecting the XY locations of individual buildings across the landscape [17]. The estimation of housing density is carried out by running a buffer operation on the point data, which converts them to polygons. These polygons serve as the spatial units in which building density is calculated. After this methodological step, both the polygon- and the point-based WUI approaches share the same set of thresholds for WUI designation which were described in the previous paragraphs.

Despite the partial methodological overlap between both WUI mapping approaches, the spatial patterns of their resulting WUI maps can differ markedly. Logically, this difference stems from their different starting points (polygon-based census data with inconsistent minimal mapping units versus coordinates of individual buildings). Consequently, it seems reasonable to expect that in regions where census data zones are small and have similar sizes, both mapping approaches will yield relatively similar WUI patterns. This is because in such cases, the zonal data will provide a discrete sampling of the continuous dataset of building locations. Yet, from my personal observations of US census data, it is clear that census blocks do not often have these characteristics, and where they do (usually in the centers of highly urbanized areas), there is no WUI due to the lack of flammable vegetation.

The differences between the two WUI mapping approaches also reflect their separate utilizations. Because it is generally unable to capture the fine-scale spatial patterns of the WUI [20], the zonal-based WUI approach is often used to provide summary statistics at coarser spatial scales (i.e., state to federal levels), which in turn enable the quantification of WUI change [7], or the identification of WUI hotspots for the purpose of strategic management decisions [21]. At the same time, these data are too coarse to pinpoint specific WUI areas in need of management, for example, to reduce fuel loads in the boundary of a specific cluster of houses. In these cases, the point-based WUI is more effective, as it essentially captures the WUI status of every single building across the landscape [14,17].

Despite its preferable traits, the biggest obstacle to broad-scale utilization of the point-based WUI method across different regions worldwide has been the lack of consistent and accessible data on building locations. Fortunately, this might change soon because recent developments in image analysis and data acquisition techniques have raised the possibility of obtaining high-resolution data on building locations across broad spatial extents. Specifically, Microsoft has generated and made publicly available a building dataset for the entire US, based on machine-learning algorithms [22] applied to recent high resolution satellite data. This dataset contains the precise footprints of 129,591,852 buildings in the entire US. The availability of these data makes it possible, for the first time, to generate a point-based WUI map for the entire US. The availability of this map allows us to address the gap in our understanding of the differences in spatial patterns between different WUI maps across large geographic extents, as it facilitates the thorough comparative analysis of resulting maps of the zonal- and point-based WUI mapping approaches, which are the two most common WUI mapping approaches today. Therefore, the main objectives of this study were to: (1) evaluate the accuracy of the Microsoft buildings dataset (hereafter the MS buildings dataset) in the context of its applicability for point-based WUI mapping across large spatial extents (an entire US state); and (2) to conduct a formal comparison of the amount and spatial pattern of the WUI generated by the zonal-based and the point-based WUI mapping approaches in that state; and, (3) to assess the contribution of different explanatory variables to the amount of spatial correspondence between the zonal-based and the point-based WUI maps. Due to the considerable computational requirements of the analysis, I restricted the spatial scope of study to the state of California. I selected

California because of its high heterogeneity of landscape and settlement configurations, in addition to it being a hotbed of the WUI fire problem in the US. In the analysis described below, I mapped the WUI from the wildfire perspective. Yet it would be straightforward to repeat the analysis in other WUI contexts.

## 2. Materials & Methods

### 2.1. Data

The study was based on three existing datasets, two of which are readily available to download for free, and one will be available soon. The first is the MS buildings dataset for the state of California, available on Github (Available online: https://github.com/microsoft/USBuildingFootprints; accessed on 27 June 2021), which contains the polygon footprints of 11,542,912 buildings that were generated using computer vision algorithms run on high resolution satellite imagery. Microsoft used a two-stage process to generate the data. First, Deep Neural Networks were used for semantic segmentation of the imagery, in order to classify pixels as parts of buildings. Then, groups of adjoining building pixels were grouped into building polygons. Most of these data are based on imagery from 2019 to 2020, but roughly one fifth of the data for California are based on earlier imagery, whose average acquisition year was 2012 (the specific imagery date of any given area is unknown). The second dataset used here is the zonal-based WUI dataset of the state of California (Figure 2), based on data from the 2010 census [7], and generated using the algorithm of [1]. This is the most comprehensive dataset available today of the WUI in the US, and it is freely available for download (https://silvis.forest.wisc.edu/data/wui-change/; accessed on 27 June 2021). The third dataset used is a prototype of the point-based WUI dataset for the conterminous US (Carlson et al. in review; available upon request) which I clipped to the state boundaries of California (Figure 2). This dataset was generated using the point-based WUI mapping algorithm [17], applied on the MS buildings dataset (described above). Given that the point-based WUI mapping approach depends on the choice of buffer distance to neighboring buildings (and there are considerable variations in spatial patterns among maps generated using different buffer distances), I obtained point-based WUI maps that were based on two buffer distances, 250 and 1000 m. In all subsequent analyses, I analyzed these two maps separately.

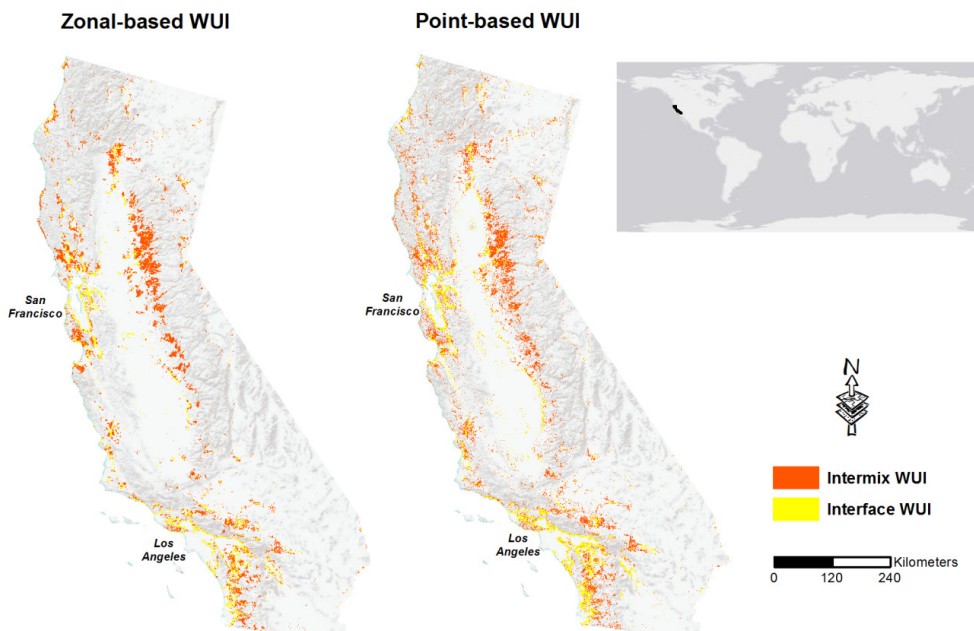

**Figure 2.** The WUI maps of California used in this study. Left: the zonal-based WUI map of California [7]. Right: a prototype of the point-based WUI of California, based on a 250 m buffer (Carlson et al. in review). Inset: the location of the study region (black) in a world map.

The varying imagery acquisition dates, coupled with data of the land cover dataset that is incorporated in WUI generation algorithms (the 2016 US NLCD), result in a slight mismatch between the dates of the two WUI products. Even though the three datasets do not represent the same date, it is assumed that they provide a valid source for comparison between the two WUI mapping approaches at the decadal scale, as well as the spatial relationship of the building data to both, because WUI development at the state scale is not very fast. Yet, some bias might be introduced into the comparison if considerable WUI development occurred after 2010, so I made the implicit assumption that WUI expansion was relatively slow. Future studies may utilize the WUI map based on the 2020 census, once it becomes available, to test if this assumption holds.

### 2.2. Preprocessing

The MS buildings dataset contains the polygonal boundaries of individual buildings. This level of detail is excessive; it greatly increases file sizes, and in turn slows down processing times. Moreover, WUI generation algorithms do not require information about the shape of buildings, but only for their location. Hence, I converted the polygon data into point data using the centroid of each building polygon as a point location. This resulted in 11,542,912 point locations for buildings in California.

The zonal- and point-based WUI datasets contain multiple land cover classes which are irrelevant for a comparative analysis between WUI patterns. To facilitate a meaningful comparison, I split both WUI products into two raster datasets: interface WUI and intermix WUI, resulting in four WUI datasets overall for the subsequent comparison steps.

### 2.3. Accuracy Assessment of the MS Building Data

Microsoft reports multiple accuracy metrics for the building footprint data at the US scale. The classification stage of the algorithm had very high accuracy levels, with pixel recall being 95.5%, and pixel precision being 94% (Github. Available online: https://github.com/microsoft/USBuildingFootprints; accessed on 27 June 2021). The polygonization stage, in which clusters of building pixels were collated into building footprint polygons, had also very high accuracy levels, with 98.5% precision and 92.4% recall. Further, the false positive ratio (non-buildings classified as buildings) was less than 1% based on a random sample of 1000 buildings across the entire dataset. Additional accuracy measures, which are not relevant for this study, reflected the accuracy of the shape-reconstruction of the building footprints, so they are not reported here. Despite the overall high accuracy of the algorithm, Microsoft did not report additional details about the accuracy assessment process, so it is not possible to estimate how well the data can serve as a basis for mapping the point-based WUI, and further, if it can be used to aid the comparison between the patterns of the point-based and the zonal-based WUI maps. Hence, it was necessary to quantify the accuracy of the building locations data prior to using them for WUI comparison purposes.

I evaluated the accuracy of the MS buildings dataset by selecting, at random, 99 building points from the entire dataset, to yield a sufficiently large sample for the purpose of the subsequent multiple regression analysis [23]. Around each building, I generated a circular sampling plot with a 250 m radius, to correspond with the buffer distance used in the generation of the finer-scale point-based WUI map used in this study (I did not use the 1000 m buffer of the second WUI map to restrict the number of houses in each sapling plot to a reasonably small number). In each sampling plot, I visually interpreted high resolution aerial and satellite photography (from Google Earth) to identify all individual buildings within the plot. In cases where visual interpretation was difficult due to high canopy cover, I used Google Street View to identify buildings from the ground level. The visually-identified buildings served as the ground truth set to which the MS buildings data are compared. The comparison between the two datasets had three possible outcomes, and for each one, I tallied the total number of buildings: (1) buildings that appeared in the MS dataset and existed in the ground truth set (true buildings); (2) appeared in the MS dataset but were not present in the ground truth set (commission errors); (3) did not appear in

the MS dataset but existed in the ground truth set (omission errors). Based on these data, I calculated two measures of classification accuracy: precision and recall. Precision denotes the fraction of correctly classified buildings out of the total number of buildings found by the MS algorithm (i.e., true positives divided by the sum of true positives and false positives). Recall denotes the fraction of correctly classified building out of all the buildings that actually exist in the plot (i.e., true positives divided by the sum of true positives and false negatives).

I then analyzed if precision and recall are affected by the number of buildings in sampling areas (the ground truth), to assess whether the accuracy of the MS buildings dataset depends on building density. Further, I quantified the relationship between each accuracy measure and the total cover of vegetated area in each sample, derived from the 2016 US National Landcover Dataset (NLCD), to test if high vegetation cover creates visual obstructions in the imagery that degrade the accuracy of the MS building identification algorithm.

### 2.4. A Comparison Between the Patterns of the Point- and Polygon-Based WUI Maps of California

To facilitate a detailed comparison between the two types of WUI products, and specifically, to assess which factors affect the correspondence between different WUI maps, I split the study area into smaller sampling units. To do that, I overlaid a rectangular grid with a spatial resolution of 10 km per square cell across a polygon that covers the entire state of California. I chose 10 km to satisfy two conditions. First, each cell should be large enough so that cells may represent variable landscapes with different amounts of WUI cover. Secondly, cells of this size are small enough so their total number across the study area is sufficiently large to facilitate a robust statistical analysis. This cell size setting resulted in a grid with 8784 cells, many of which were irrelevant because they intersected non-land areas, or occurred where there are no WUI areas (by both WUI mapping approaches). I therefore deleted from this grid all cells that did not overlap any WUI area, resulting in a grid with 4780 cells. The remaining cells had varying levels of the four possible WUI types/classes: point-based WUI interface or intermix (based on the 250 m point-based WUI map), and zonal-based interface or intermix. The process was repeated for the point-based WUI interface and intermix that were based on the 1000 m buffer distance. To facilitate the comparison between WUI types at the cell level, I quantified the area of the zonal-based and the point-based interface and intermix WUI in each cell. I also calculated a spatial agreement index between each corresponding pair of WUI maps (zonal vs. point interface, and zonal vs. point intermix, at two buffer distances) using intersection over union (IoU, also known as Jaccard index). IoU denotes the amount of intersecting WUI areas in both maps, divided by the union of WUI areas in those maps. It reaches one when the WUI zones in both maps are perfectly aligned, and drops to zero if there is no overlap at all. I used IoU as the dependent variable in subsequent statistical models that attempt to explain the degree of spatial correspondence between WUI patterns for pairs of maps generated by the zonal-based and the point-based WUI approaches at two buffer distances.

To account for the potential effect of building density on the correspondence between different WUI types, I extracted the number of MS buildings per cell. This number varied greatly, from zero to 103,465. Beyond the number of buildings per cell, it is also possible that the correspondence between different WUI maps is affected by the spatial pattern of building locations. To quantify it, I first created, for each cell, a binary raster map with a 30 m resolution in which "1" denotes pixels that intersect a building, and "0" denotes pixels that do not. This raster is a discrete representation of the spatial pattern of building locations, and as such it facilitates the usage of various landscape metrics to quantify the spatial configuration of settlements across the landscape. Based on this raster, I quantified the percent cover of buildings (total area of "1" pixels divided by total cell area, i.e., 100 km$^2$) as a simple measure of building coverage; and calculated the aggregation index [24] to quantify the tendency of buildings to be spatially aggregated within a given sampling cell. The aggregation index is the number of like adjacencies (pixels of "1" that are adjacent to other "1" pixels) divided by the maximum possible number of like adjacencies (when all

"1" pixels are aggregated). Finally, to account for the possibility that the size distribution of US census blocks (which underlies the generation of the zonal-based WUI) affects the correspondence between the two WUI products, I calculated, in each sampling cell, the mean and the standard deviation of the areas of the census blocks that were defined as interface or intermix WUI in the zonal-based WUI map. A flowchart that depicts the above steps appears in Figure 3.

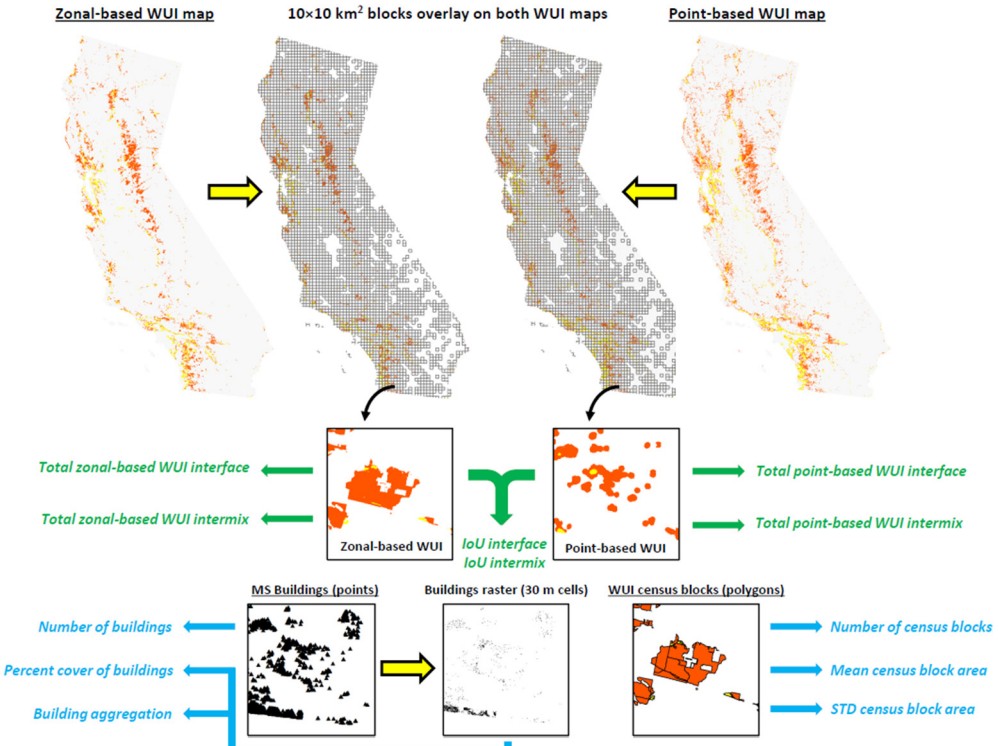

**Figure 3.** A flowchart depicting the generation of variables for statistical analyses. Existing data product names are underlined. Names in green italics are dependent variables in the analysis, and names in blue italics are predictor variables. Arrows connect between variables and their data source.

I quantified the effect of the above variables on the correspondence between WUI maps generated by either approach (point-based at two buffer distances vs. zonal-based) using two statistical models that were fit in R [25]. To compare differences in WUI area between mapping approaches (and for interface WUI and intermix WUI separately), I used linear models with the difference between WUI areas (point-based minus zonal-based) as the dependent variable, and all of the variables described in the previous paragraph as independent variables. Given that in the majority of cases, the point-based WUI area was larger than zonal-based WUI area, and to improve interpretability, I omitted from the analysis cases where the zonal-based WUI area was larger than the point-based area. Consequently, positive effect sizes for model variables reflect predictors that decrease the agreement between mapping approaches, and negative effects size imply improved agreement.

To compare the IoU values between WUI maps, I used generalized linear models with a quasi-binomial error distribution. The IoU at the respective buffer distances was the dependent variable, and the variables described above were the predictors. In models based on spatial data such as these, spatial autocorrelation is a cause of concern. I therefore evaluated if models were affected by spatial autocorrelation by visual inspection of empirical variograms of model residuals, generated in R using the gstat package [26,27], and found no evidence for such effect.

## 3. Results

### 3.1. Accuracy of the MS Building Dataset

The validation dataset of 99 sample sites contained 16,561 true buildings overall (compared to 16,028 buildings in the MS dataset for the same sites), with individual sites containing from 0 (a site with eight buildings in the MS dataset, all of which do not exist in reality) to 381 buildings, and a mean of 167.28 buildings per site. In general, the accuracy of the MS buildings dataset was very high, in line with the accuracy measures reported by Microsoft. Mean precision was 98.4%, and mean recall was 95.4%, implying that the algorithm's commission error rate is very small, while its ability to identify the true number of buildings in a given site is very high (i.e., a very small omission error). The correlation between the numbers of buildings identified by the MS algorithm to the ground truth was extremely high, 0.99, but the slope of the linear regression between the numbers of ground truth buildings and the buildings in the MS dataset was significantly larger than one (β = 1.03 ± S.E. 0.008, $p < 0.001$), implying a tendency of the MS algorithm to slightly under-estimate the number of buildings in high-density areas (Figure 4). Both precision and recall were not significantly explained by the combination of the amount of vegetation cover in a site, or the number of actual buildings in a site (generalized linear models with a binomial error distribution), further implying that the MS algorithm works well in different landscape settings. The fact that neither building density nor vegetation cover affect the accuracy of the MS algorithm, makes it likely that it will perform well in the task of identifying buildings in the WUI, where building density tends to be low and vegetation cover tends to be high. Moreover, the overall high accuracy of the MS dataset makes the building numbers variable extracted from it a suitable predictor in the subsequent statistical models that compare the patterns of the point-based and zonal-based WUI maps.

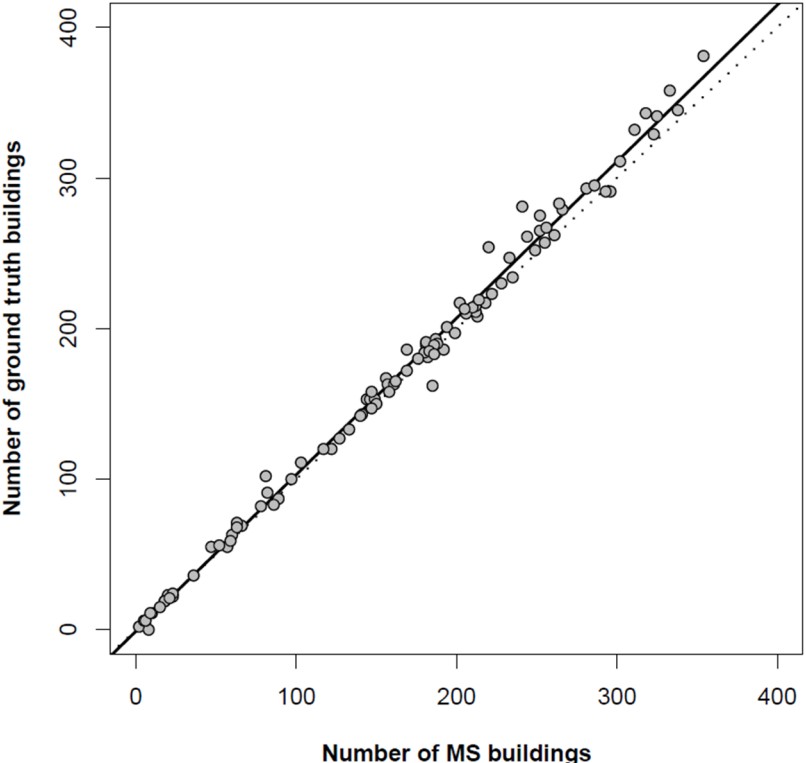

**Figure 4.** The relationship between the number of ground-truth buildings and the number of buildings in the MS dataset, across 99 sampling areas of 250 m radius. The dotted line is the 1:1 slope with zero intercept, whereas the solid line depicts the slope of a simple linear model fit to the data.

### 3.2. WUI Area in the Zonal- and Point-Based WUI Maps of California

The total coverage and the spatial patterns of the zonal-based and the point-based WUI maps were compared across 4780 rectangular cells of 100 km$^2$ in California, for interface and intermix WUI separately. Of these, 4629 cells contained intermix patches in at least one of the two WUI maps (at 250 m buffer distance), and 2623 cells contained interface patches in at least one map (same buffer distance). When the point-based WUI was based on a 1000 m buffer distance, these numbers decreased to 3002 and 2091 for intermix and interface WUI, respectively. The total area designated as interface WUI was 804,634.9 ha in the zonal-based map, vs. 1,323,210 and 1,525,677 ha in the point-based maps at 250 and 1000 m buffer distances, respectively. The total area of intermix WUI in both map types was considerably larger, 1,897,923 ha in the zonal-based map, vs. 2,624,238 and 4,106,519 ha, in the point-based maps at 250 and 1000 m buffer distances, respectively. Hence, in both WUI types, the point-based mapping approach consistently yields a much larger WUI area, and, specifically in point-based maps, a larger buffer distance results in a much larger WUI at the state level. Moreover, the point-based WUI mapping approach predicted considerably larger WUI areas in each sampling cell compared to the zonal-based approach (Figure 5), as the slopes of the linear relationships between the predicted areas by the zonal vs. the point-based approach were significantly smaller than one at both 250 m buffer distances (Figure 5a—Interface WUI: effect size = 0.6 (S.E. $\pm$ 0.005), $p < 0.001$; Figure 5b—Intermix WUI: effect size = 0.95 (S.E. $\pm$ 0.006), $p < 0.001$), and 1000 m buffer distances (Figure 5c—Interface WUI: effect size = 0.47 (S.E. $\pm$ 0.007), $p < 0.001$; Figure 5d—Intermix WUI: effect size = 0.6 (S.E. $\pm$ 0.006), $p < 0.001$). Yet, the fit between both approaches in the case of intermix WUI was better (the slope coefficients were closer to one), compared to the fit of the interface WUI. Additionally, the fit was better when the point-based WUI was generated using a shorter buffer distance (effect sizes plus/minus standard errors did not overlap, and were closer to one at 250 m buffer distance).

Across all four mapping combinations (interface and intermix, at 250 and 1000 m buffer distance), an increase in the number of buildings tended to significantly increase the difference between the amount of area predicted by the point-based approach compared to the zonal-approach (Figure 6; the figure also includes the standardized effect sizes and the standard errors). A similar, albeit weaker effect, was found for the degree of building aggregation across cells, where cells with high aggregation had larger difference in WUI area between mapping approaches. On the other hand, an increase in the variation in the size of census blocks contributed to smaller differences in the WUI area between approaches (significant in all cases except for interface WUI at 250 m buffer). Larger census blocks tended to significantly increase difference in WUI areas between maps, but only in the case of intermix WUI (Figure 6, bottom row). Regardless, the amount of explained variation (multiple R$^2$) in the difference in WUI area was relatively small across these models: 28.5% (interface/250 m), 15.8% (interface/1000 m), 9.3% (intermix/250 m), and 23% (intermix/1000 m).

### 3.3. Spatial Patterns of the WUI in the Zonal- and Point-Based WUI Maps of California

The spatial correspondence between the zonal-based and the point-based (with 250 m buffer distance) interface WUI maps (i.e., their IoU) was significantly affected by multiple characteristics of the landscape in the sampling cell (Figure 7, top-left panel), yet the amount of deviance explained in the interface WUI model was rather low, 0.08. The IoU of interface maps was negatively affected by the standard deviation of the area of census blocks defined as interface and the percent cover of buildings in the cell. IoU tended to increase when the mean area census blocks defined as interface WUI was larger, and when buildings tended to be aggregated across the sample cell.

The IoU between maps calculated using point-based WUI data at 1000 m buffer distances was affected by a slightly different combination of predictors (Figure 6, top-right panel), and the model had twice as much deviance explained (0.16). Specifically, IoU increased when cells contained more census blocks, when census blocks had varying sizes,

when the number of buildings increased, and when buildings were more aggregated. IoU decreased when the percent cover of buildings in the cell increased. The main differences between the results at 250 and 1000 m buffer distances are the switching sign of the significant effect of the standard deviation of the size of census blocks (negative at 250 m, positive at 1000 m), the addition of the number of census blocks as a significant predictor at 1000 m, and the omission of the mean area of census blocks as a significant predictor at 1000 m. In general, correspondence between maps of interface WUI, regardless of buffer size, was better when buildings tended to be aggregated and covered smaller percentages of the cell. Yet, there is no clear geographic pattern in these areas of California (Figure 8).

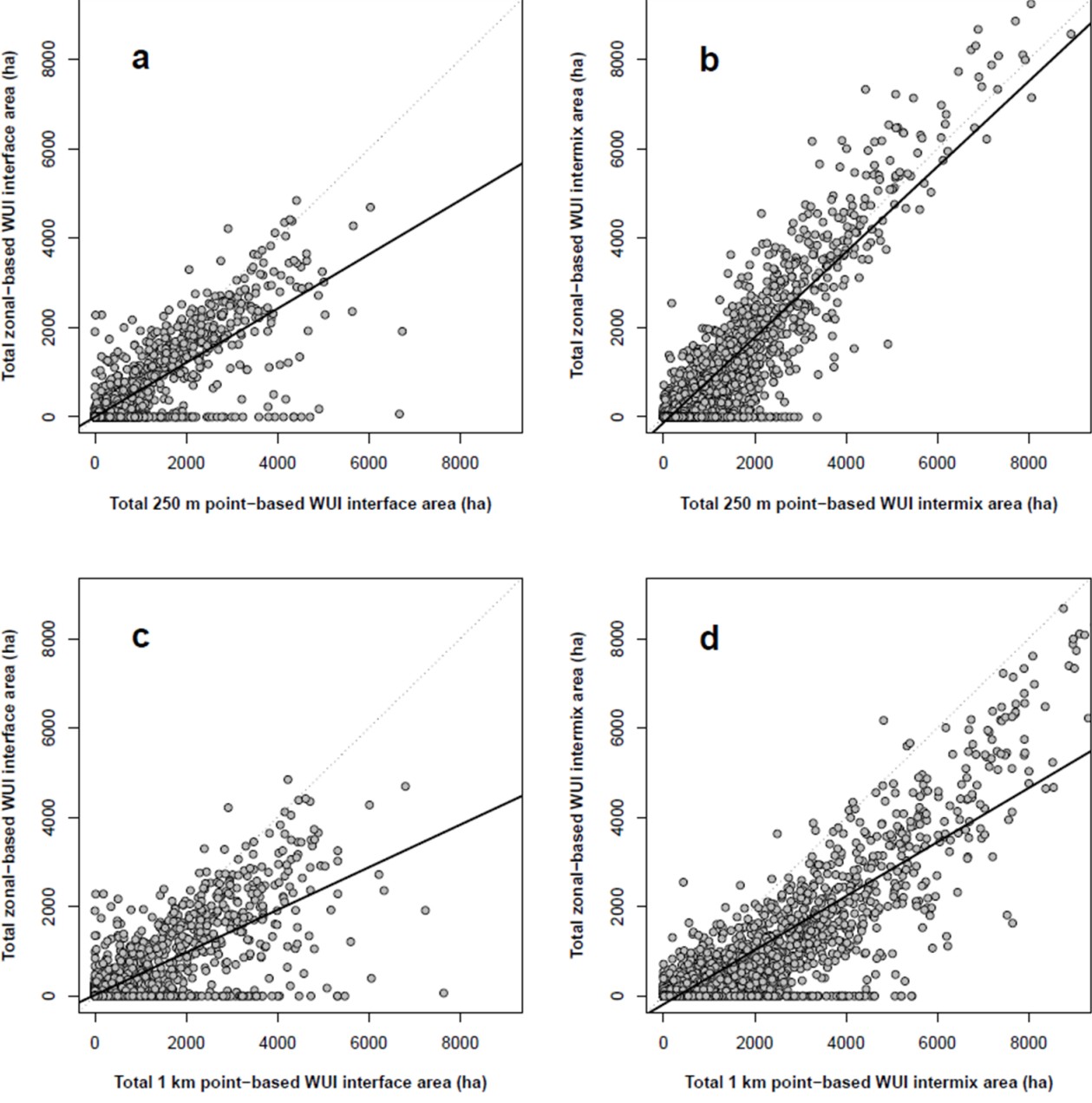

**Figure 5.** Relationships between the total area of interface (panels **a**,**c**) and intermix WUI (panels **b**,**d**) as quantified by the zonal-based (Y axes) and the point-based (X axes) WUI mapping approaches. In the top row, point-based WUI is based on a 250 m buffer, whereas in the bottom row, it is based on a 1 km buffer. In all panels, the dotted lines depict a 1:1 relationship between axes, whereas the solid lines depict the slope of a simple linear model fit to the data.

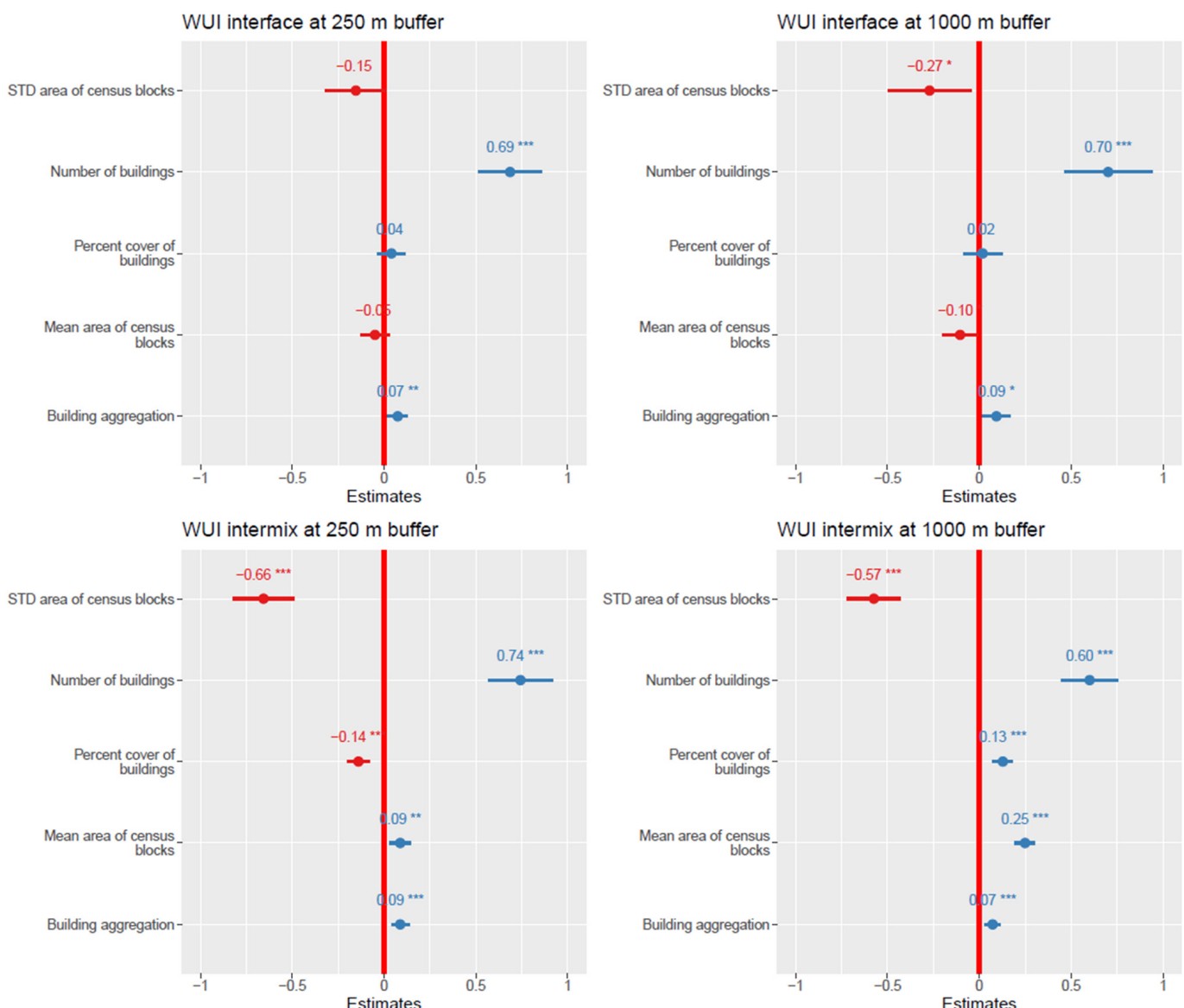

**Figure 6.** Standardized effect sizes of linear models for the correspondence in terms of area (point-based minus zonal-based) of zonal-based vs. point-based WUI maps. The top row is for interface WUI; whereas the bottom row is for intermix WUI. The left panels depict results of models with point-based WUI generated a 250 m buffer distance, whereas the right panels are based on 1000 m buffers. The thick red line highlights an effect of zero; a predictor whose standard error overlaps it has a non-significant effect WUI area differences. Significance levels: *** ($p < 0.001$), ** ($p < 0.01$), * ($p < 0.05$). Model terms are explained in Section 2.4.

In contrast to the case of interface WUI, the correspondence between maps of intermix WUI (where point-based WUI was based on a 250 m buffer distance) was much better explained by the combination of multiple landscape features (Figure 7, bottom-left panel), as the amount of deviance explained in the IoU model of intermix WUI was 0.43 (compared to 0.08 for interface). The IoU of intermix maps was positively affected by: the number of intermix patches, the standard deviation of the areas of the census blocks behind them, the percent cover of buildings across the cell, and the amount of aggregation of buildings. The number of buildings in the sampling cell had a negative effect on the IoU of intermix maps. The results of the corresponding model for point-based WUI at 1000 m buffer distance (Figure 4, bottom-right panel) were qualitatively the same (i.e., the amount of explained deviance, 0.43, was almost the same; the same variables were significant; and effect sizes were similar in magnitude and had the same direction). Hence, for the sake of brevity, I will

not describe them here. In general, though, the correspondence between maps of intermix WUI tended to improve in areas with not many buildings, but where these buildings tended to be aggregated in clusters across large parts of the landscape at low densities. In terms of geographic location, these areas tended to be located in the mountain regions of central California, specifically in the Sierras (Figure 8).

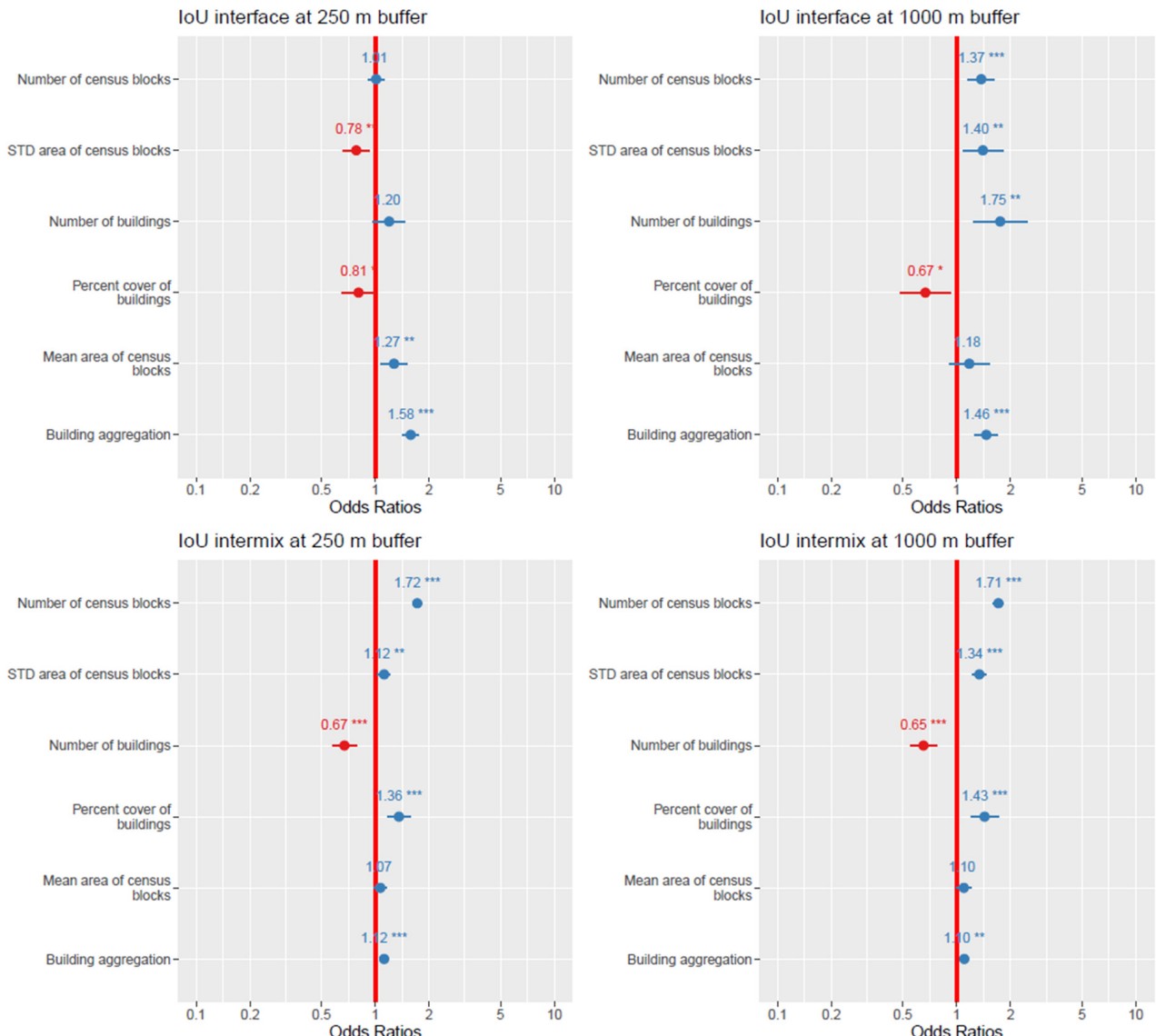

**Figure 7.** Effect sizes (as odds ratios) of the GLMs for the correspondence (as IoU) between the spatial patterns of zonal-based vs. point-based WUI maps. The top row depicts interface WUI, and the bottom row depicts intermix WUI. The left panels depict results of models of point-based WUI (at 250 m buffer distance), whereas the right panels are based on point-based WUI at 1000 m buffer distance. Coefficients are standardized, and the thick red line highlights an odds ratio of one. Significance levels: *** ($p < 0.001$), ** ($p < 0.01$), * ($p < 0.05$). Model terms are explained in Section 2.4.

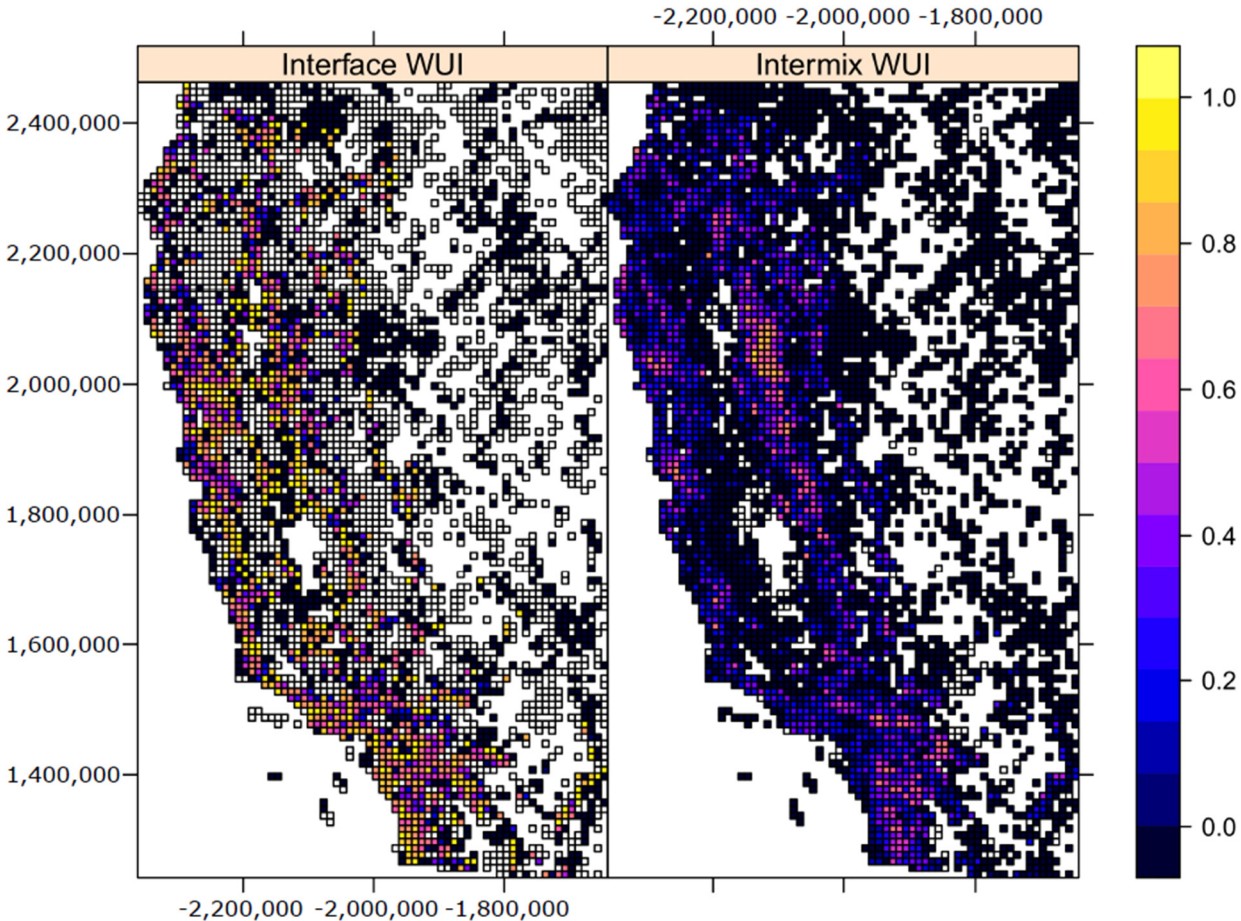

**Figure 8.** The spatial patterns of the Intersection over Union (IoU) measure of correspondence between zonal-based and point-based WUI maps. Left: a comparison of interface WUI maps; right: a comparison of intermix WUI maps. Rectangles depict the 10-by-10 km sample cells used in the comparison. Empty areas denote locations where there were no interface or intermix WUI areas in both mapping approaches.

## 4. Discussion

In this study, I found that the spatial correspondence between WUI patterns generated by two of the most common WUI mapping approaches, the zonal-based and the point-based, is low-to-moderate at best, and inconsistent across space. Point-based WUI maps tend to designate considerably larger areas as part of the WUI (whether interface or intermix), and the spatial extent of California's WUI is much larger according to the point-based approach. Moreover, the specific parameterization of the point-based WUI approach (in terms of the choice of buffer distance at the first stage of the generation process) has considerable implications on both the area and the spatial patterns of the resulting WUI. These two issues have profound implications for policy and management purposes, as they suggest that resources to mitigate environmental problems in the WUI, specifically fire, should be spread across much larger areas that were considered before [1,7,21].

The findings of this study, which are first and foremost relevant to California, highlight that the point-based WUI map tends to consistently define much larger areas as WUI compared to the zonal-based WUI map. Why is that so? The results of the statistical analysis offer the following clues. The number of buildings in a sample plot had the largest positive effect on the difference between maps, followed by building aggregation (i.e., differences in WUI area between point-based WUI and zonal-based WUI maps were larger in cells with more buildings, and when they tended to be aggregated across space). For intermix WUI, larger census blocks also contributed to this difference. This result is in line with my initial expectation that smaller census blocks provide a finer-scale representation of

the spatial pattern of buildings across space, which may serve as a discrete approximation to the continuous pattern of building effect in the point-based WUI approach. As for variables that increased agreement between mapping approaches, the only variable that had a consistent effect is the standard deviation in the area of census blocks. Such cases exist mostly in exurban and rural areas, which are more likely to be defined as WUI in the first place (in contrast to dense urban areas, where census blocks tend to have less size variation; while at the same time, are less likely to be defined as the WUI). Another pattern, the increase in the total area of the WUI in point-based WUI maps when a larger buffer distance is used, is unsurprising. This is because in a given sampling block with a regular building arrangement, larger buffers create disproportionally larger spatial units (zones) which are then designated as WUI, as long as building density remains above the threshold of 6.17 per square km [17].

In general, one crucial aspect to consider when comparing the patterns of WUI maps is that despite profound differences in their patterns, there is no one "true" WUI map. This is because WUI maps differ in definitions, contexts, and purposes [5,12,19]. Hence, the considerable differences found here between the outcomes of different mapping approaches should not be viewed as a recommendation to use a given mapping approach at the expense of the other. Rather, they highlight the need to ensure that the choice of mapping approach aligns with its intended purpose. For example, coarse scale mapping that attempts to provide basic statistics about WUI dynamics across time may rely on the zonal-based approach, because it is easier to implement, requires data that are less labor (or computationally) intensive to create and update, and provide reasonable estimate of WUI extent which can facilitate comparison across states or countries [7]. In contrast, if the aim of the analysis is pinpointing specific locations for management actions (i.e., for fuel treatments to reduce flammability or fire spread rate near settlements), then the point-based WUI is the better approach as it accounts for the specific location of individual buildings [14,17,28]. Yet, the issue becomes even more complicated if one considers other WUI processes besides wildfire as the mapping objective [3], such as the case of invasive species [29,30]. To usefully map the WUI in terms of the risk of spread of invasive species, it is no longer sufficient to classify wildland areas based on a flammability criterion (and to then apply some threshold value to determine if a given spatial unit is part of the WUI). Instead, the WUI needs to be identified according to a combination of the particular characteristics of multiple species at once, each with its own preference to a different set of environmental conditions (which in turn are manifested by different land cover types). The resulting WUI map is likely to differ considerably among different mapping approaches, and one might ask which map should be preferred. If the WUI designation is related to issues of safety or conservation, then one might prefer to err on the side of caution, by selecting the mapping approach that maximizes WUI area.

Unfortunately, the choice to map the WUI according to one approach or the other does not rely solely on their purpose. This is due to issues of data availability and limitations. In many countries, the only available data on the spatial distribution of settlements are at the zonal level (e.g., the US census, CORINE land cover maps in the EU); hence, it is only possible to apply the zonal-based WUI mapping approach in them. In other places, data on individual building locations are available [14,17,18], which facilitates the usage of the point-based WUI approach. Yet, these data are often available at small spatial extents (e.g., a single municipality), or are non-available to the research community due to issues of property rights and data privacy. Hence, data limitations are possibly the main driver of the choice of WUI mapping approaches worldwide, and on the characteristics of the resulting map products (i.e., point-based WUI maps tend to cover much smaller spatial extents compared to zonal-based maps). It remains to be seen if high-resolution data on building locations, similar to the MS-dataset used here, will become increasingly available worldwide, and crucially, if these data will be updated on a regular basis in order to facilitate the important task of mapping WUI dynamics.

## 5. Conclusions

The results of this study highlight the significant differences between the spatial patterns of WUI maps generated by the two most common WUI mapping approaches today. These differences are profound both in terms of overall WUI coverage and its spatial configuration across landscapes. These incongruences also have important implications for management and policy decisions aimed at reducing fire risk in the WUI, as they suggest that the overall effort required to manage fire in the WUI (which corresponds with total WUI coverage), together with the spatial locations that require mitigation (which reflect the spatial configuration of the WUI), will vary between mapping approaches. Hence, to successfully identify areas-at-risk of wildfire, in order to reduce fire risk in the WUI, there is need to develop means to resolve differences between maps and mapping approaches, and to effectively integrate the information about the spatial patterns of the WUI that is deduced from these maps.

**Funding:** This research has been granted funding from the European Union's Horizon 2020 research and innovation programme under the Grant Agreement no. 101003890.

**Institutional Review Board Statement:** Not applicable.

**Informed Consent Statement:** Not applicable.

**Data Availability Statement:** The zonal-based WUI map used in this study is available for download from: http://silvis.forest.wisc.edu/data/wui-change/ accessed on 27 June 2021. The MS building dataset is available for download from: https://github.com/microsoft/USBuildingFootprints. accessed on 27 June 2021. For other data inquiries, please contact the corresponding author.

**Acknowledgments:** I thank Amanda Carlson and Volker Radeloff for helpful discussions that greatly improved the focus of this study, and for providing the point-based WUI map used in this study.

**Conflicts of Interest:** The author declares no conflict of interest.

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
