# Peer review of "A Comparative Analysis of Two Major Approaches for Mapping the Wildland-Urban Interface: A Case Study in California"

_land, doi:10.3390/land10070679_

Round 1

Reviewer 1 Report

General comments: 

The submitted paper is an interesting approach to the analysis of wildland-urban interface and the way it can be mapped. The methodology design should be more justified, and it should be based on scientific reasoning and not on personal choices. Once there is a well-grounded methodology, there will be significant results. In addition, there are still some lacks in the presentation of the manuscript, which would need extra amendments.   

Structure of the manuscript 

Title 

The title is proper for the presented study. I would suggest removing the dash that separates the two sentences. 

Abstract 

I found the abstract adequate and concise.  

Introduction 

Lines 42-43: Please, try to rewrite this sentence in a more formal way. “(but see other approaches in [12,15] which build up- 42 on different concepts for WUI delineation)”. 

Line 49: in the case of the fire-centric WUI. 

Please, state clearly which is the scientific gap for this study. 

Material and methods 

Please, include a chart or diagram showing the main steps of your methodology. 

Please, include figure or figures displaying the study area. 

Line 190: Why did you select 99 building points? Why exactly 99? This needs a justification. 

Line 192: Why did you select 250 m for the radius? Why exactly 250 m? This needs a justification. 

Line 219: Why did you select this grid of exactly 10 x 10 km? This needs a justification. 

Line 225: Why did you select 250 m for the buffer? Why exactly 250 m? This needs a justification. 

Results 

Once there is a well-grounded methodology, there will be significant results. The results are too related with the personal choices of the author and how he conducted the methodology. 

Conclusions 

Please, include a conclusions section. 

Author Response

Response to reviewer 1 comments

Title 

The title is proper for the presented study. I would suggest removing the dash that separates the two sentences. 

Response: I replaced the dash with a colon to emphasize the different parts of the sentence.

Abstract 

I found the abstract adequate and concise.  

Introduction 

Lines 42-43: Please, try to rewrite this sentence in a more formal way. “(but see other approaches in [12,15] which build up- 42 on different concepts for WUI delineation)”. 

Response: I rewrote the sentence to make it more fluid

Line 49: in the case of the fire-centric WUI. 

Response: I emphasized this in the revised sentence

Please, state clearly which is the scientific gap for this study. 

Response: I added the following statement to the end of the background section: "The availability of this map allows us to address the gap in our understanding of the differences in spatial patterns between different WUI maps across large geographic extents, as it facilitates the thorough comparative analysis of resulting maps of the zonal- and point-based WUI mapping approaches, which are the two most common WUI mapping approaches today"

Material and methods 

Please, include a chart or diagram showing the main steps of your methodology. 

Response: this chart already appears in Figure 1. A schematic that depicts all subsequent steps of the analysis will be cumbersome and difficult to understand, so I'd rather not add it to the manuscript.

Please, include figure or figures displaying the study area. 

Response: Done. See new Figure 2.

Line 190: Why did you select 99 building points? Why exactly 99? This needs a justification. 

Response: I wanted to use a sufficiently large sample size to establish the statistical relationship in the data. The number itself is arbitrary, and is suitable as long as it's large enough. 

Line 192: Why did you select 250 m for the radius? Why exactly 250 m? This needs a justification. 

Response: the 250 m radius corresponds with the parameter of the point-based WUI map used in this study. I explain this in the revised text.

Line 219: Why did you select this grid of exactly 10 x 10 km? This needs a justification. 

Response: I wanted to use a sufficiently large area for map comparisons, which will also generate a sufficiently large sample size to establish the statistical relationship between the maps. The number itself is arbitrary, and any alternative number is suitable as long as it satisfies the above criteria. 

Line 225: Why did you select 250 m for the buffer? Why exactly 250 m? This needs a justification. 

Response: the 250 m buffer simply reflects the corresponding point-based WUI map which was presented in the data section (2.1.). I rephrased the text to better reflect that.

 Results 

Once there is a well-grounded methodology, there will be significant results. The results are too related with the personal choices of the author and how he conducted the methodology. 

Response: I hope that with the clarified methodology, the results will be more significant to the reviewer.

Conclusions 

Please, include a conclusions section. 

Response: I added a conclusion section to the revised manuscript.

Reviewer 2 Report

I feel that the topic is interesting but there are some points to be revised.

  1. I feel that the figure 1 is about research process. How about moving into the methods part?
  2. LAND is the journal of social science. How about making the section of literature reviews.
  3. The more detailed information of the study area with the map is needed.
  4. More explanation about data is needed. I feel that table about materials is beneficial for easy understanding.
  5. Conclusion section is needed for summary and the direction of future studies.

Author Response

Response to reviewer 2 comments

I feel that the figure 1 is about research process. How about moving into the methods part?

Response: Thanks for your suggestion, but the current location in the background is correct because the figure reflects existing approaches.

LAND is the journal of social science. How about making the section of literature reviews.

Response: I'm afraid I didn't understand this comment. The background section contains a thorough review of the relevant literature.

The more detailed information of the study area with the map is needed.

Response: I added a new figure, (Figure 2) which depicts the study area.

More explanation about data is needed. I feel that table about materials is beneficial for easy understanding.

Response: the explanation of specific data and methods is expanded following specific comments by reviewer 1. There are no materials here, so I'm afraid I can't tell which table should I add.

Conclusion section is needed for summary and the direction of future studies.

Response: Thanks for this suggestion. I added a conclusion section to the revised manuscript.

Reviewer 3 Report

Please, see my comments in the attached file. Thank you!

Author Response

Response to reviewer 3 comments

Thanks for the paper, which I read with great interest. Indeed, comparative analysis of two major approaches for mapping the wildland-urban interface is a very high-potential area of research. Thus, mapping the WUI is required to identify areas-at-risk. The article describes the methods qualitatively and makes valuable conclusions. Results from profound implications for policy and management purposes, suggest that resources to mitigate environmental problems in the WUI, specifically fire, should be spread across much larger areas than researches considered before.

Response: Thank you for your positive assessment!

However, I would like to clarify a few points.

The author could expand the literature review. For example, the author emphasized that “WUI has grown in past decades, especially in the US…” It seems to me not so correct sentence. Although many articles are published in English, there is an amazing body of literature published in other languages (e.g., French, Spanish, Portuguese) related to wilderness and urbanization problems. WUI was faster developed all over the World, for example, in China (Cao, Y et al, Wenyu Jiang et al, Lin, S. et al, etc), Russia (V. Bocharnikov, Eugene Egidarev, etc), Europe (Christoph Plutzar et al, etc), UK (Carver et al, etc) and many others. Even though the paper concerns only California, a reference to the experience of scientists from other countries would be valuable.

Response: Thank you for your suggestion. I added several more references to the background section. Notice, though, that the original background section did include references to studies across the entire world, and was not restricted to the study area.

The detailed explanation of the terms in Figure 1 is probably missing. Please, more detailed explain what exactly you mean is “interface WUI” and “intermix WUI”.

Response: corrected accordingly. I now describe these terms in the caption.

Concerning figures 3 and 5, please, explain what exactly is “STD area census blocks”.

Response: this term is explained in lines 268-269 in the methods section. I now added a pointer in the caption to lead the reader to the explanation of terms in section 2.4.

Overall, the study makes a positive impression. It is recommended for publication after minor corrections. The paper is fascinating to read. I wish the author every success in further research.

Response: thank you for your kind evaluation!

Round 2

Reviewer 1 Report

The author did not address the main concerns of the methodology:

1. Why did you select 99 building points? Why exactly 99? This needs a justification.

2. Why did you select this grid of exactly 10 x 10 km?This needs a justification.

The results of the study are influenced by these 2 main sources. The author's reply for these 2 concerns are not sufficient for a high level international paper. Please, justify it with a sufficient and statistical scientific basis.

Author Response

Response to reviewer comments

The author did not address the main concerns of the methodology:

Why did you select 99 building points? Why exactly 99? This needs a justification.

Response: I apologize if my explanation wasn't sufficiently clear. There are many rules of thumb for choosing sample size in statistics. The simplest one, for a multiple regression model, is having at least 10 samples per predictor. In my case, there are (at most) 6 predictors, so a reasonable sample size to ensure statistical power would be 60. Any number above that would suffice, and really, differences in values will be arbitrary. A more thorough rule of thumb is based on Green (1991), which suggests a sample size larger than: 50 + 8*predictors, which in my case is 98. Hence my choice of 99 is in line with this rule as well.  Just in case, I clarified this in the revised text. I hope that this satisfies your criticism.

2. Why did you select this grid of exactly 10 x 10 km?This needs a justification.

Response: here, too, sample size (thousands) is much larger than the minimally recommended size for a multiple regression (99). Hence from a statistical perspective, all is good. At the same time, I had to consider the meaning of the results at this grain size (10 km). As the WUI is typically assessed at the landscape scale, a single block cannot be too small. 10 km is large enough to represent a landscape unit with varying levels of land cover, and WUI coverage. I respectfully repeat my assertion that the specific number (10km) is rather meaningless as long as it adheres with the two conditions I detailed above. Hence from an interpretation standpoint, there isn't much difference between 9km, 10km, or 10.6km. All of these are sufficiently large landscape units which yield a sample size that far exceeds the statistical requirement of the regression analysis.

The results of the study are influenced by these 2 main sources. The author's reply for these 2 concerns are not sufficient for a high level international paper. Please, justify it with a sufficient and statistical scientific basis.

Response: I appreciate your concern and hope that my explanations are satisfactory.

Reviewer 2 Report

Honestly speaking, the author did not follow my comments. I cannot understand the process diagram in the introduction section. I feel the literature section is needed but the author did not make it. In addition, more detailed map is needed. I still cannot know where the site is? The map in world level is needed. I cannot follow the materials the author said. Table of the materials is needed. But author did not follow the advice. Major revision is required.

Author Response

Response to reviewer comments

Honestly speaking, the author did not follow my comments. 

Response: I apologize if my previous responses were unclear. In some instances (e.g., the literature section), I simply didn't understand the comment. I other cases, I made additional changes accordingly. I hope that my responses below will clarify things.

I cannot understand the process diagram in the introduction section. 

Response: this is not a process diagram, but a flowchart that explains the existing methodology of WUI mapping. Regardless, I added a process diagram in a new Figure (3). Hopefully, it will make the methods easier to understand.

I feel the literature section is needed but the author did not make it.

Response: I'm afraid that I still don't understand this comment. The introduction section contains a thorough literature review, including references to 22 published studies, which cover every aspect of this manuscript. What else is missing here? Notice that the journal's formatting instructions do not mention a separate literature section.

In addition, more detailed map is needed. I still cannot know where the site is? The map in world level is needed. 

Response: I agree with this point an expanded figure 2 to include a world map. The location of the study region is marked on it.

I cannot follow the materials the author said. Table of the materials is needed.

Response: there are no materials in the text, just variables. The newly added figure 3 provides a graphical representation of all variables.